# Extended State Observer Based-Backstepping Control for Virtual Synchronous Generator

**Shamseldeen Ismail Abdallah Haroon** [1,2] , **Jing Qian** [1,*], **Yun Zeng** [1] , **Yidong Zou** [3] **and Danning Tian** [4]

1   Faculty of Metallurgical and Energy Engineering, Kunming University of Science and Technology, Kunming 650093, China
2   Department of Electrical and Electronic Engineering, College of Engineering Science, Nyala University, Nyala 63311, Sudan
3   School of Power and Mechanical Engineering, Wuhan University, Wuhan 430000, China
4   Academic Affairs, School of Global Public Health, New York University, Long Island City, NY 11101, USA
*   Correspondence: qj0117@kust.edu.cn; Tel.: +86-137-0844-0678

**Abstract:** The penetration of distributed generators (DGs)-based power electronic devices leads to low inertia and damping properties of the modern power grid. As a result, the system becomes more susceptible to disruption and instability, particularly when the power demand changes during critical loads or the system needs to switch from standalone to a grid-connected operation mode or vice versa. Developing a robust controller to deal with these transient cases is a real challenge. The inverter control method via the virtual synchronous generator (VSG) control method is a better way to supply the system's inertia and damping features to boost system stability. Therefore, a nonlinear control strategy for VSG with uncertain disturbance is proposed in this paper to enhance the system stability in the islanded, grid-connected, and transition modes. Firstly, the mechanical equations for a VSG's rotor, which include virtual inertia and damping coefficient, are presented, and the matching mathematical model is produced. Then, the nonlinear backstepping controller (BSC) method combined with the extended state observer (ESO) is constructed to compensate for the uncertainty. The Lyapunov criteria were used to prove the method's stability. Considering the issue of uncertain items, a second-order ESO is built to estimate uncertainty and external disruption. Finally, the suggested control strategy is validated through three simulation experiments; the findings reveal that the proposed control method has an excellent performance with fast response and tracking under various operating situations.

**Keywords:** renewable energy source; inverter; microgrid; virtual synchronous generator; backstepping technology; extended state observer

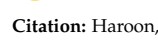



## 1. Introduction

At the present, the limitation of fossil fuels, the growth in environmental pollution caused by the combustion of fossil fuels in the energy production sector, and technological innovations have become global concerns [1,2]. In addition to the modern vision of distribution networks, which aims to replace the old pattern that depends on the large central sources of energy delivered to the consumer through transmission networks, small networks generate electricity closer to customer areas based on renewable energy [3]. As a result of the need for a more efficient generation method to avoid the effects mentioned above, renewable energy technologies and distribution generation have emerged [1–3]. The world is witnessing a rapid expansion in renewable energy sources, as well as a larger share of electrical energy produced from renewable sources as grid-connected or isolated systems, which makes the operation and stability control of distributed generation units and their grid connectionsparamount [4]. Small networks are an electrical system for transferring electrical energy from multiple types of DGs connected together to form a microgrid, and they provide a way to integrate these DGs into the power grid, which is developed with

the technology of renewable energies and distributed generation [4,5], where the power electronic devices play a very important role in this discipline [6]. As is well known, DGs based on power electronic devices, such as solar panels, wind turbines, fuel cells, and so on, have low inertia and damping properties, which have an impact on system stability and dynamic performance, making the system more susceptible to perturbations [5]. Due to the lack of grid inertia and damping support, the stable operation of the power grid has been challenged as the penetration of power electronic devices based on DGs into the power grid has increased. In order to solve these issues, the virtual synchronous generator, where the power electronic inverter is controlled to replicate the properties of traditional synchronous generators, was proposed as a promising strategy [6–8].

Many researchers and academics have attempted to overcome the abovementioned challenges by modeling the inverter power component and adding rotor swing equations to offer virtual inertia, allowing the inverter to work as a VSG [6,8–10]. Several configurations for VSG systems have been introduced worldwide since 2008 [6,11]. Recently, various control methods have been applied for VSG to overcome the microgrid stability and ensure a smooth transition during disruption, such as particle swarm optimization, including the voltage angle deviations of generators, which was suggested in [12], oscillation damping method of VSG using pole placement, proposed in [13], a hybrid control method of the inertia and the damping island microgrids [14,15], imitation excitation control method [16], improved fuzzy logic controller [17–21], and so on. Moreover, many ideas focused on modifying and adapting the droop control methods. In [22], an improved virtual synchronous generator control technique based on adaptive droop coefficient addressed the problem of low power distribution accuracy and large frequency oscillation in the island microgrid. The proposed approach displayed excellent power distribution accuracy, which improved the power grid system's dynamic performance and frequency stability. Even more, the voltage deviation can be increased or decreased using the VSG control approach to improve the voltage stability of microgrid systems. By modeling VSG based on self-adaptive control, the effects of the inertia coefficient and the droop coefficient on voltage stability were explored in [23]. Self-adaptive control of the droop coefficient can reduce the adjusting time and the voltage deviation during the disturbance and transient conditions. To enhance the dynamic performance of the microgrid during the system transfer from islanding mode to grid-connected mode, a virtual impedance-based VSG control approach was introduced in [24]. In [25], fixed-parameter damping methods of VSG control using state feedback to solve low-frequency oscillation and enhance operational power ripple attenuation capabilities by using a low-pass filter were given.

Furthermore, from the perspective of providing frequency stability, inertial, and damping support, an adaptive virtual inertia control strategy based on an improved bang-bang control strategy to improve the frequency stability of the system for a microgrid was presented in [26]. In addition, the work in [27] studied a frequency systematic control approach employing a virtual synchronous generator and an expanded whale algorithm to identify the perfect solution of control parameters. Despite the fact that VSG output characteristics are identical to those of a classical synchronous generator, the system stability during transient situations, such as sudden increasing power demand, transition process, and the off-grid operation mode, is as yet troublesome. However, in the VSG microgrid connection association with tracking the phase voltage, amplitude, and frequency, an additional measurement device, phase closed loop (PLL) is required, limiting the critical output of the distributed power generation system [24,28]. The pre-synchronization control approach for grid connection of a VSG was published in [28], and it introduces rotational inertia and damping via droop control to take part in grid frequency regulation and provide a smooth transition at the moment of grid coupling. The frequency controller and excitation controller were both designed by developing the VSG model and controller. Other than whatever has been referenced, there are numerous uncertainties and significant external disruptions in the scenario of the practical implementation of the VSG, making the design of control methods difficult. BSC approaches can be used to manage systems

with significant levels of nonlinearity, which piqued many researchers' interest, such as the following: a high-performance controller and disturbance rejection can be achieved using a backslapping control approach with a disturbance observer based on stability theory [29], and an integral sliding mode control approach and backstepping control were suggested for VSG in [30] to eliminate transient oscillations. An adaptive control filter backstepping method was designed to solve the low system inertia and support the grid frequency for the system containing different distributed generators that could provide power stability and frequency enhancement [8], which is more effective than conventional control approaches for avoiding the power system instability and frequency optimization obtained in conditions of disruptions and uncertainty in system parameters. In [31], an adaptive sliding mode control methodology based on VSG electromagnetic transient properties was developed to solve the distributed power system's low inertia and damping problem and improve system stability.

Nevertheless, ESO was presented for tracking order states and dynamic errors in non-linear systems with unknown disturbances [32–37]. An active power controller for smooth power tracking for a grid-connected VSG was established based on linear active disturbance rejection control to deal with power oscillations in [34]. The findings demonstrate that the grid-connected VSG has excellent power control performance. Moreover, the developed control system can promptly transmit active power to the grid without overshooting the power reference, and there is no requirement for PLL when the grid frequency fluctuates. As mentioned earlier, the existing VSG control solutions have successfully resolved some stability issues caused by low inertia and damping properties. However, the effects of the large current generated during the transition process, which causes the microgrid instability and oscillations in the output power, are not considered. As a result, developing a new control strategy for a microgrid system to minimize significant current deviation during transient conditions while maintaining system output parameters, including active power, voltage, and frequency constant, is very important. Thus, this paper focuses on overcoming these issues without needing a pre-synchronous device. The following are the main contributions of this study:

1. On the basis of the previous investigations, the basic mathematical model of VSG is established according to the extent of the situation.
2. With the addition of compensating signals, a nonlinear BSC based on ESO is constructed for the virtual synchronous generator system, enabling system stability in off-grid mode, grid-tied mode, and transition operation.
3. ESO is developed to estimate unknown disturbances and ensure tracking of dynamic system errors based on the external disturbances in the system model, allowing the microgrid to operate similarly to the actual operation.

The remainder of this study is laid out as follows: Section 2 presents the system structure and modeling, followed by the proposed control method in Section 3, and the corresponding simulation results are presented in Section 4. Section 5 discusses the simulation results of the proposed method compared with other control methods. Finally, Section 6 elaborates the conclusions of this paper.

## 2. System Structure and Modeling

### 2.1. Mathematical Model of Synchronous Generator

The synchronous generator's voltage output generated in the stator and armature windings [34,38], can be written as

$$V = E - R_s i - L_s \frac{di}{dt} \tag{1}$$

where $V$ is the voltage at the terminal, $E$ is an electromotive force produced by excitation, and $L_s$ and $R_s$ are inductance and resistance, respectively, which represent armature impedance elements. The term induced electromotive force must satisfy

$$E = M_f i_f \dot{\theta} \sin \tilde{\theta} - M_f \frac{d i_f}{dt} \cos \tilde{\theta} \tag{2}$$

where $M_f$ is the rotor and the stator winding's mutual inductance, $i_f$ is the current of excitation, $\psi_f$ is the rotor's flux, and $\dot{\theta} = \omega$ is the generator output angular frequency. The mathematical equation of the synchronous generator according to the Euler–Newton concept [8,30,38], can be written as

$$\dot{\omega} = \frac{1}{J} (T_m - T_e - D\Delta\omega) \tag{3}$$

In this equation, $J$ is the moment of inertia of rotating components, $T_m$ is the mechanical torque, $T_e$ is the electromagnetic torque, $D$ is the damping coefficient mostly produced by rotor damping winding, and $\Delta\omega = \omega - \omega_0$, $\omega_0$ is the reference angular frequency of equal $2\pi f$.

### 2.2. VSG Proposed Structure

The most basic form of an inverter circuit that can be utilized in microgrids and controlled as a virtual synchronous generator to provide inertia and damping features is shown in Figure 1. The inverter's DC side is fed by a DC voltage source that indicates the output voltage of a renewable energy source or a battery via a DC link. The grid is connected to the inverter output via an LCL filter. The electronic part of the inverter circuit represents the control system, which is used to control the power part. This circuit is clearly equivalent to the electrical section of the synchronous generator from a technical standpoint, where the neutral point voltage $e_{abc} = \begin{bmatrix} e_a & e_b & e_c \end{bmatrix}^T$ describes the three-phase bridge arm voltage, which represents the synchronous generator internal potential, and $e_a$ is the internal potential in phase $a$. The stator winding impedance of the synchronous generator is represented by $L_o$ and $R_o$ on the inverter side; thus, the impedance components of the rotor winding are described by $L_f$ and $R_f$. The output voltage of inverter (capacitor voltage) $u_{abc} = \begin{bmatrix} u_a & u_b & u_c \end{bmatrix}^T$ indicates the synchronous generator's three-phase terminal voltage, $u_a$ is phase $a$ terminal voltage, and the inductance current $i_{abc} = \begin{bmatrix} i_a & i_b & i_c \end{bmatrix}^T$ can be used to describe the output current of the synchronous generator, where $i_a$ is the current through phase $a$.

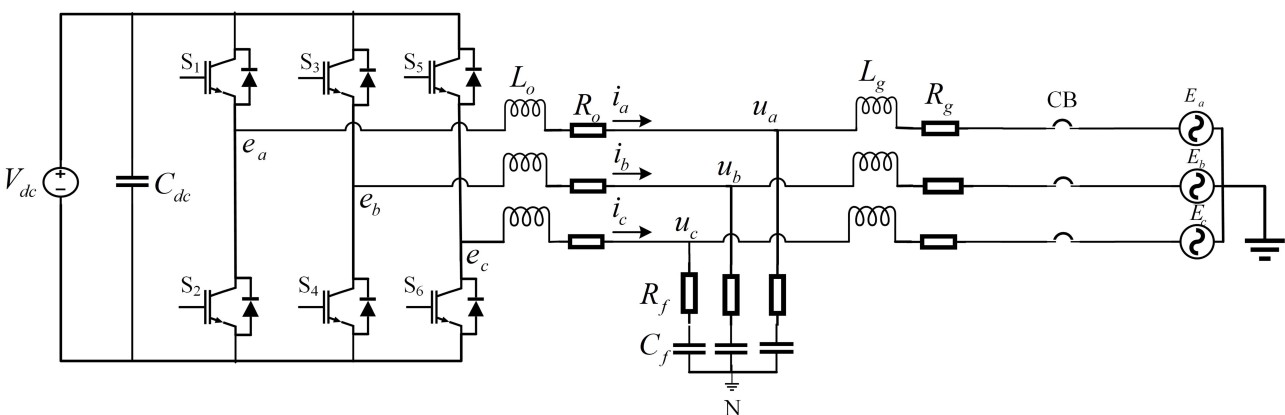

**Figure 1.** Power part of the inverter unit.

It is known that the LCL filter output causes reactive power loss; therefore, $u_{abc}$ and $i_{abc}$ will takevirtual synchronous generator output. According to the instantaneous power

theory, the active and reactive power output can be determined using the electromotive force, and the inductance current [8,10,38], as

$$P_e = 1.5\psi_f i\omega \cos\delta \tag{4}$$

$$Q_e = 1.5\psi_f i\omega \sin\delta \tag{5}$$

where $\delta = \theta - \varphi$, which represents the power angle. From Equation (4), the electrical torque can be derived:

$$T_e = \frac{P_e}{\omega} = 1.5\psi_f i_{abc} \cos\delta \tag{6}$$

Based on Equation (1), which represents the output voltage of the synchronous generator, the electromagnetic equation of virtual synchronous generator can be expressed as

$$u_{abc} = e_{abc} - L_o\frac{di_{abc}}{dt} - R_o i_{abc} \tag{7}$$

where $u_{abc}$ is VSG terminal voltage, $R_o$ is virtual armature resistance; $L_o$ is the virtual armature inductance, and both of them represent the components of the inverter output impedance, while $i_{abc}$ is output current signal, and therefore

$$i_{abc} = i\cos\widetilde{\varphi} \tag{8}$$

where $\varphi$ is current angle. According to Equation (2), the part $e_{abc}$ can be expressed as

$$e_{abc} = \psi_f\omega\sin\widetilde{\theta} \tag{9}$$

in the above equation, $e_{abc}$ is the virtual internal potential, $\psi_f$ is the virtual rotor flux which represents the virtual exciter output, and $\theta$ is the phase angle. Based on the voltage regulation equations, the expression for the reactive voltage and power droop control loop function, as shown in equation form, is as follows:

$$V_{ref} = V_{rat} - k_q\left(Q_{ref} - Q_e\right) \tag{10}$$

where $V_{ref}$ is the system voltage reference value, $V_{rat}$ is rated voltage, $k_q$ is the droop coefficient of voltage reactive power, $Q_{ref}$ is the reference command for reactive power, and $Q_e$ is the actual reactive power output. Knowing that, the magnetic flux control circuit, as depicted in Figure 2 below, is responsible for generating the voltage amplitude in real synchronous generators.

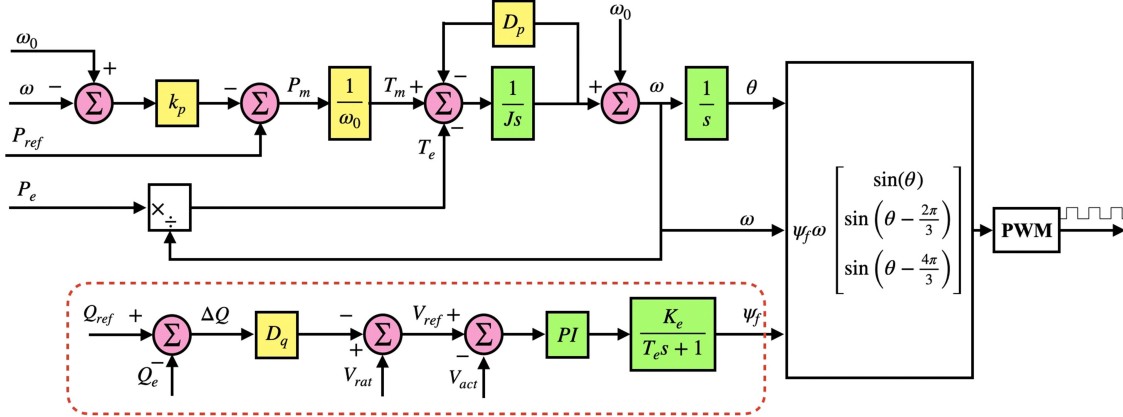

**Figure 2.** VSG basic control structure.

Figure 2 shows a basic structure of the VSG control system, including the flux control circuit; it can be seen that the error generates by comparing the reference value of the

voltage $V_{ref}$ generated by the reactive power voltage in the droop control loop with the amplitude of the feedback measured output voltage $V_{rat}$. The error is sent to the virtual exciter after flowing through the PI controller. A low-pass filter is applied in virtual synchronous generators [8,24] for the following reasons:

- To mimic the synchronous generator's flux weakness caused by the DC voltage applied to the rotor's excitation circuit and the inductance of the rotor coil, which causes the stator to delay.
- A low-pass filter with a time constant can remove and filter high-frequency components of the output signals, which helps with the design of the backstepping controller based on ESO.

The design method is described in the next section.

## 3. Proposed Control Method

This section presents a backstepping controller with combined ESO, along with an analysis and a detailed design process based on the VSG core equations noted in the previous section.

### 3.1. Voltage Control Strategy

To design and integrate the backstepping controller into the flux control loop, the error signal generated by the droop control function links with the backstepping controller output signal, as depicted in Figure 3, which shows that the low-pass filter input signal is formulated as follows:

$$u = u_1 + u_2 \tag{11}$$

where $u_2$ is droop control function, and $u_2$ is the proposed output controller as compensator signal. The transfer function of the filter with a time constant has the following expression:

$$G_f(s) = \frac{\psi_f}{u_2} = \frac{K_e}{1 + t_e s} \tag{12}$$

where $k_e$ is the gain of the low-pass filter, and $t_e$ is the low-pass filter time constant.

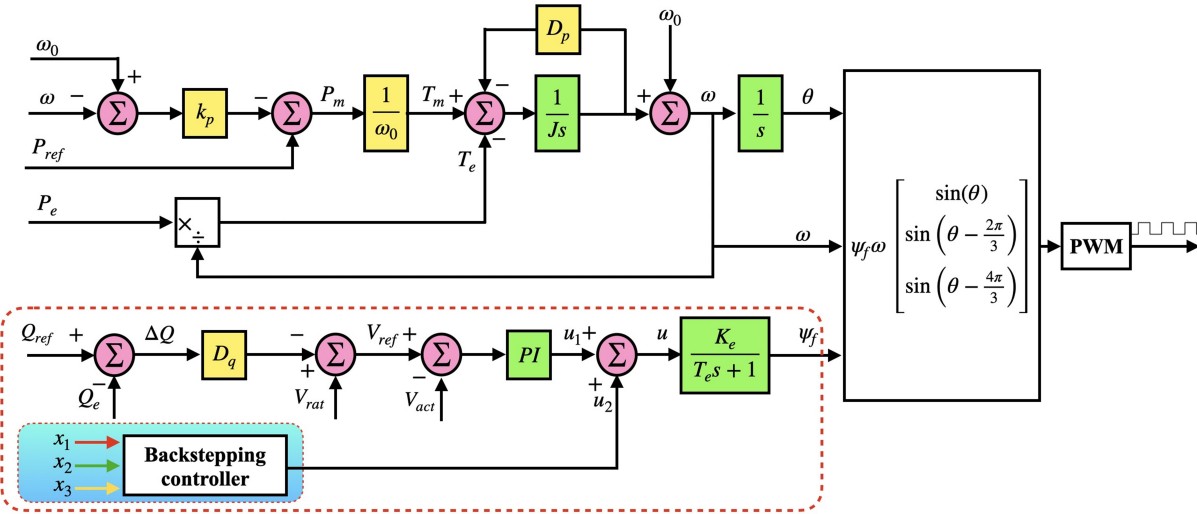

**Figure 3.** Structure of VSG control circuit with compensating signal.

The dynamic change of power angle $\Delta\delta$, angular frequency $\Delta\omega$, and electrical torque $\Delta T_e$ are the system state variables that can be obtained as

$$
\begin{cases}
\Delta\dot\delta = \Delta\omega, \\
\Delta\dot\omega = -\frac{D_P}{J}\Delta\omega - \frac{1}{J}\Delta T_e, \\
\Delta\dot T_e = \frac{3}{2}i\psi_f\cos\Delta\delta - \frac{3}{2}i\psi_f\Delta\omega\sin\Delta\delta + \frac{1}{t_e}\Delta T_e + (\frac{3k_e}{2t_e}i\cos\Delta\delta)u_2 + d.
\end{cases}
\tag{13}
$$

where $a_1 = -\frac{D_P}{J}, a_2 = -\frac{1}{J}, a_3 = \frac{3}{2}i\psi_f, a_4 = -\frac{3}{2}i\psi_f, a_5 = \frac{1}{t_e}, a_6 = \frac{3k_e}{2t_e}i, b = a_6\cos x_1$, and $d$ represents uncertain bounded disturbance.

The model of the system (13) is simplified to execute the recommended control method by assuming that $x_1 = \Delta\delta, x_2 = \Delta\omega$, and $x_3 = \Delta T_e$, which represents the system state variables. As a result, the following equation are used to summarize the model (13):

$$
\begin{cases}
\dot x_1 = x_2, \\
\dot x_2 = a_1 x_2 + a_2 x_3 \\
\dot x_3 = a_3\cos x_1 + a_4 x_2\sin x_1 + a_5 x_3 + bu_2 + d.
\end{cases}
\tag{14}
$$

The system model (14) is a system with a bounded disturbance, d, that is unknown. A backstepping control strategy with second-order ESO is described and constructed in the section below to ensure the system's stability and tracking performance. The system tracking errors can be defined by

$$
e = \begin{cases}
e_\delta = x_1 - x_r, \\
e_\omega = x_2 - c_\delta, \\
e_{T_e} = x_3 - c_\omega.
\end{cases}
\tag{15}
$$

where $e_\delta, e_\omega$, and $e_\delta$ indicate the differences in variables between the reference and actual values, and $x_r$ is the reference value of $x_1$.

1. The rotor angle dynamic error can be defined as

$$
\dot e_\delta = x_2 - \dot x_r \tag{16}
$$

The Lyapunov function is defined as follows to stabilize the virtual rotor angle $\delta$:

$$
V_1 = \frac{1}{2}e_\delta^2 \tag{17}
$$

The derivative of selected function is given as follows:

$$
\dot V_1 = e_\delta\dot e_\delta = e_\delta(e_\omega + c_\delta - \dot x_r) \tag{18}
$$

In order to meet the system stability requirement, the appropriate power angle virtual control law $c_\delta$ is designed as follows:

$$
c_\delta = -k_\delta e_\delta + \dot x_r \tag{19}
$$

The following equation is generated after substitution and incorporation:

$$
\dot V_1 = -k_\delta e_\delta^2 + e_\delta e_\omega \tag{20}
$$

where $-k_\delta e_\delta^2 \le 0$, and $k_\delta$ is a positive constant.

2. The mechanical angular velocity dynamic error can be presented as

$$
\dot e_\omega = a_1 x_2 + a_2 x_3 - \dot c_\delta \tag{21}
$$

The Lyapunov function should be chosen as follows to stabilize the angular velocity:

$$V_2 = V_1 + \frac{1}{2}e_\omega^2, \tag{22}$$

then, the following derivative can be obtained:

$$\dot{V}_2 = -k_\delta e_\delta^2 + e_\omega(e_\delta + a_1 x_2 + a_2 x_3 - \dot{c}_\delta) \tag{23}$$

According to Equation (15), the relation between electrical torque dynamic error and the state variable $x_3$ can be described:

$$x_3 = e_{T_e} + c_\omega \tag{24}$$

By substituting into Equation (23), the following result can be obtained:

$$\dot{V}_2 = -k_1 e_\delta^2 + e_\omega(e_\delta + a_1 x_2 + a_2(e_{T_e} + \alpha_2) - \dot{c}_\delta) \tag{25}$$

To satisfy the stability condition, the angular velocity virtual control law $c_\omega$ must be developed in the following way to satisfy the initial condition:

$$c_\omega = \frac{1}{a_2}[-e_\delta - a_1 x_2 + \dot{c}_\delta - k_\omega e_2], \tag{26}$$

then, the following result can be obtained:

$$\dot{V}_2 = -k_\delta e_\delta^2 - k_\omega e_\omega^2 + a_2 e_\omega e_{T_e} \tag{27}$$

where $k_\delta$ and $k_\omega$ are real numbers; thus, the quantity $-k_\delta e_\delta^2 - k_\omega e_\omega^2 \le 0$.

3. The electrical torque dynamic error can be defined as

$$\dot{e}_{T_e} = \dot{x}_3 - \dot{c}_\omega \tag{28}$$

The Lyapunov function $V_3$ is chosen as

$$V_3 = V_2 + \frac{1}{2}e_{T_e}^2 \tag{29}$$

The derivative of function $V_3$ can be defined as

$$\dot{V}_3 = -k_\delta e_\delta^2 - k_\omega e_\omega^2 + a_2 e_\omega e_{T_e} + e_{T_e}(a_3 \cos x_1 + a_4 x_2 \sin x_1 + a_5 x_3 + b u_2 + d - \dot{c}_\omega) \tag{30}$$

To satisfy the stability condition, similarly, the term $\dot{V}_3$ should be equal to or less than zero; thus, the virtual controller must be designed as follows:

$$u_2 = \frac{1}{b}[-k_{T_e} e_{T_e} - a_3 \cos x_1 - a_4 x_2 \sin x_1 - a_5 x_3 - a_2 e_2 - \hat{d} + \dot{c}_\omega], \tag{31}$$

then, the following result can be obtained:

$$\dot{V}_3 = -k_\delta e_\delta^2 - k_\omega e_\omega^2 - k_{T_e} e_{T_e}^2 \tag{32}$$

where $k_\delta$, $k_\omega$, and $k_{T_e}$ are positive constants; thus, the quantity $-k_\delta e_\delta^2 - k_\omega e_\omega^2 - k_{T_e} e_{T_e}^2 \le 0$. The structure of the designed controller is shown in Figure 4.

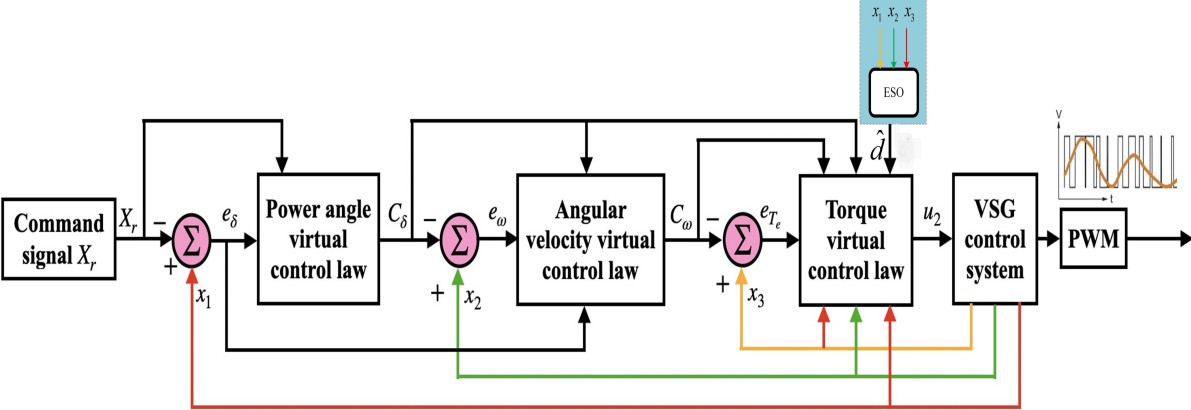

**Figure 4.** Structure of backstepping controller combined ESO.

### 3.2. ESO Observer

In the above, Equation (31) contains externally uncertain disturbance, d. Thus, to ensure the system's stability and to estimate the unpredictable external disruption, the nonlinear second-order ESO is designed as follows:

$$\begin{cases} e_1 = z_1 - x_3, \\ \dot{z}_1 = z_2 - \beta_1 e_1 + a_3 \cos x_1 + a_4 x_2 \sin x_1 + a_5 x_3 + b u_2 - \hat{d}, \\ \dot{z}_2 = -\beta_2 fal(e_1, \alpha, \tau). \end{cases} \tag{33}$$

where $z_i (i = 1, 2)$ is the observation of the state variable in the system (14), $\beta_1$ and $\beta_2$ are observer gain settings, and the function $fal(e_1, \alpha, \tau)$ is given by

$$fal(e_1, \alpha, \tau) = \begin{cases} \frac{e_1}{\tau^{\alpha-1}} & |e_1| \leq \tau, \\ |e_1|^\alpha sign(e_1) & |e_1| > \tau. \end{cases} \tag{34}$$

where $e_1$ is the observer error, and $\alpha$ is a constant in the interval (0, 1). Furthermore, $\tau$ is the length of the interval of the linear segment [32,37,39]. To ensure the stability of the observer described by Equation (33), the following assumptions must be satisfied:

**Assumption 1.** *When $e_1$ satisfies the condition $|e_1| \leq \tau$, then the function $fal(e_1, \alpha, \tau) = \frac{e_1}{\delta^{\alpha-1}}$, then*

$$\begin{cases} \dot{e}_1 = e_2 - \beta_1 e_1, \\ \dot{e}_2 = -\beta_2 \frac{e_1}{\delta^{\alpha-1}} - \dot{d}. \end{cases} \tag{35}$$

*When the system goes into the steady state, the derivative of $\dot{e}_1$ and $\dot{e}_2$ is almost close to zero:*

$$\begin{cases} \dot{e}_1 = e_2 - \beta_1 e_1 = 0, \\ \dot{e}_2 = -\beta_2 \frac{e_1}{\delta^{\alpha-1}} - \dot{d} = 0. \end{cases} \tag{36}$$

*Result: when the part $|\beta_2| > |\dot{d}.\delta^{\alpha-1}|$ is established, it can be obtained that $z_1$ is close to $x_3$, and $z_2$ is close to d.*

**Assumption 2.** *When $e_1$ satisfies the condition $|e_1| > \tau$, then the function $fal(e_1, \alpha, \tau) = |e_1|^\alpha sign(e_1)$, and*

$$\begin{cases} e_1 = -\frac{\dot{d}\delta^{\alpha-1}}{\beta_2}, \\ e_2 = -\beta_1 \frac{\dot{d}.\delta^{\alpha-1}}{\beta_2}. \end{cases} \tag{37}$$

When the system goes into the steady state, the derivatives of $\dot{e}_1$ and $\dot{e}_2$ are almost close to zero.

$$\begin{cases} \dot{e}_1 = e_2 - \beta_1 e_1 = 0, \\ \dot{e}_2 = \pm \beta_2 |e_1|^\varepsilon - \dot{d} = 0. \end{cases} \tag{38}$$

Result: Likewise, when the part $|\beta_2| > |\dot{d}|$ is formed, it can be obtained that $z_1$ is close to $x_2$, and $z_2$ is close to $d$.

From the above analysis, we conclude that the stability of the system based on low-order ESO can be ensured when the following conditions are satisfied: $|\beta_2| > |\dot{d}|$ and $|\beta_2| > |\dot{d}.\delta^{\alpha-1}|$. For more equations details, please see Appendix A.

## 4. Simulation Results

This section presents simulation results for the VSG-controlled inverter to demonstrate the suggested methodology's practicality. For verification and validation of the suggested control strategy, the model is developed in the MATLAB/Simulink environment R2018a. In order to move closer to the actual system operation situation, the simulation experiments are conducted for three cases based on the microgrid configuration of Figure 1. The cases include two types of load variation disturbances and transition processes (grid connection and disconnection); the simulation time is 1.2 s for each case. The parameters for the system model during the simulation tests are chosen as listed in Tables 1 and 2. Moreover, based on the Lyapunov stability theory, the appropriate $k_\delta$, $k_\omega$, and $k_{T_e}$ are selected to achieve the rapid stability of the system as follows: $k_\delta = 30$, $k_\omega = 40$, and $k_{T_e} = 0.1$. Furthermore, the parameters for the second-order extended state observer were selected as follows: $\beta_1 = 100$, $\beta_2 = 100$, $\alpha = 0.5$, and $\tau = 0.1$

**Table 1.** Microgrid parameters.

| Parameter | Value | Description |
|-----------|-------|-------------|
| $V_{dc}$ | 800 V | DC-link voltage |
| $E_g$ | 400 V | Line-ground voltage |
| $R_g$ | 0.065 Ω | Grid resistance |
| $L_g$ | 0.001 mH | Grid inductance |
| $R_o$ | 5 Ω | Output resistance |
| $L_o$ | 0.5 mH | Output inductance |
| $C_f$ | 10 μf | Filter capacitor |
| $R_f$ | 2 Ω | Filter damping resistor |
| $F_{sw}$ | 10 kHz | Switching frequency |

**Table 2.** The parameters of the VSG controller.

| Parameter | Value | Description |
|-----------|-------|-------------|
| $P_{ref}$ | 10 kW | Rated active power |
| $J$ | 0.5 kg·m$^2$ | Moment of inertia |
| $D$ | 20 | Damping coefficient |
| $\omega_0$ | 314 rad/s | Nominal angular frequency |
| $k_p, k_q$ | 0 .0001, 0.001 | Frequency and voltage droop gain |
| $k_e, t_e$ | 1, 0.1 | LPF time constant and gain |
| $k_p, k_i$ | 2, 2000 | PI controller parameters |

### 4.1. Load Variation Disturbances

The variation of the load demand influences the microgrid output, so the microgrid control shall interact with such disturbances to provide stable operation and acceptable

performance. The simulation process in these cases is based on Figure 1, which operates in an isolated operation mode.

### 4.1.1. Case-1

At t = 0 s, the inverter provides active power to the supplied 7 kW local load. Then, when t = 0.4 s, the load suddenly increased to 10 kW; at t = 0.8 s, the 3 kW is removed. The simulation results of this case are shown in the following figures: Figure 5 demonstrates the active power variation result, Figure 6 shows the frequency response, and Figures 7 and 8 depict the variation of three-phase voltage and current, respectively. The system tracking errors are illustrated in Figure 9.

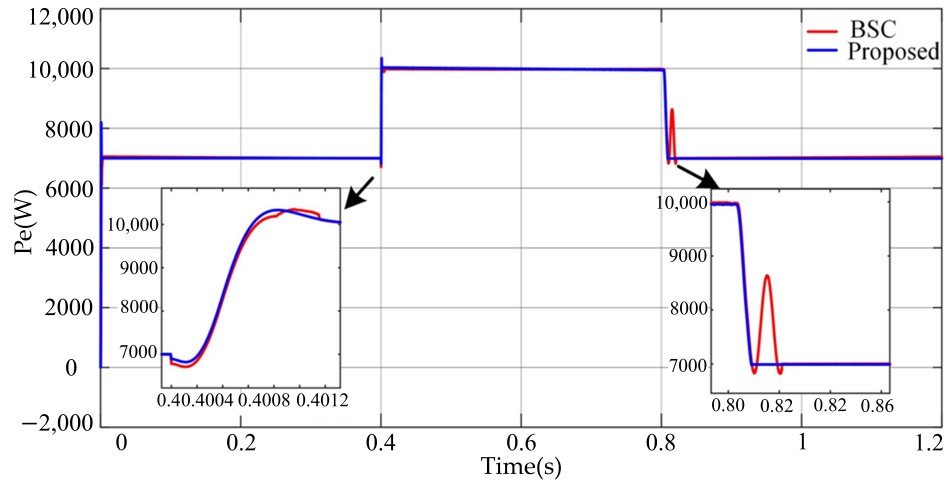

**Figure 5.** Active power variation result of case-1.

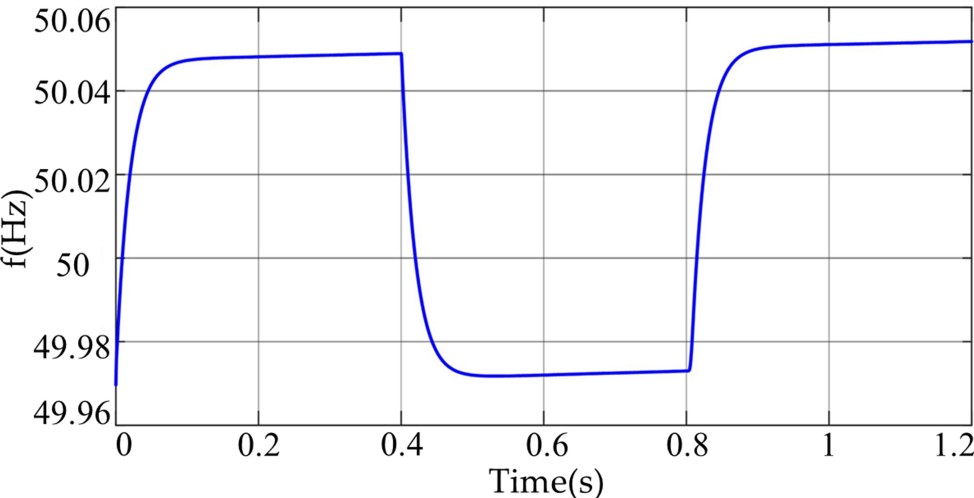

**Figure 6.** Frequency variation result of case-1.

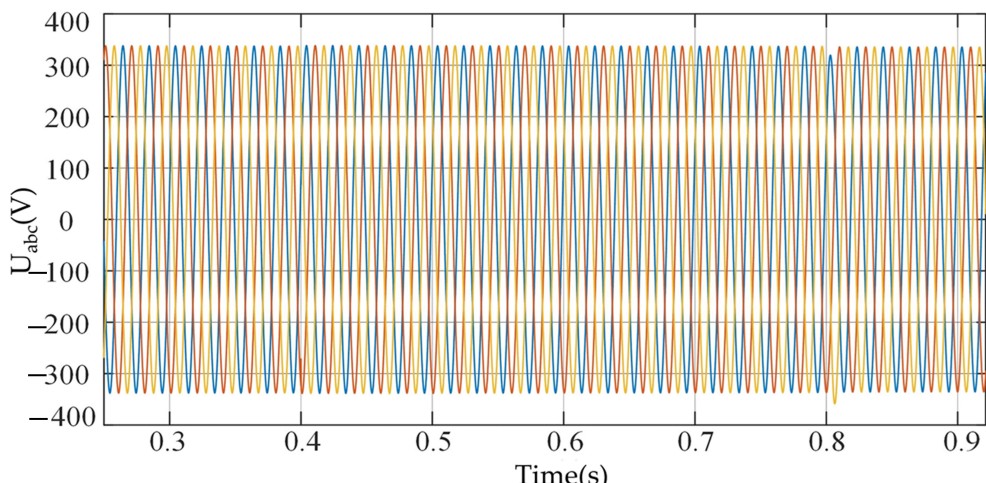

**Figure 7.** Variation of three-phase voltage in case-1.

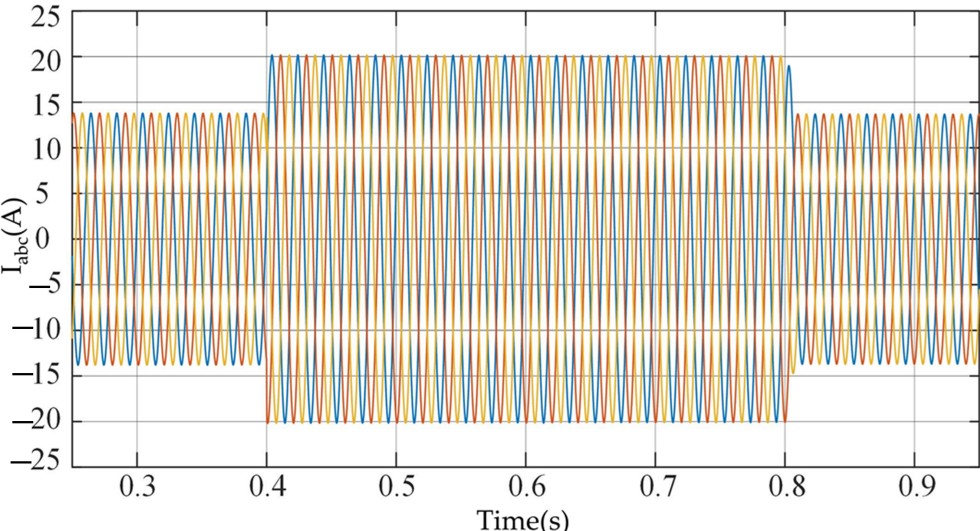

**Figure 8.** Variation of three-phase current in case-1.

4.1.2. Case-2

In this case, the system operates stably and supplies 10 kW load demand initially. At t = 0.4 s, 3 kW active power was removed; then, when t = 0.8 s, 3 kW load was added again. Similarly, the simulation results of this case are illustrated as follows: Figure 10 depicts the active power variation result, Figure 11 shows the frequency variation result, Figure 12 illustrates the three-phase voltage situation, Figure 13 exhibits the three-phase current variation, and Figure 14 depicts the system tracking errors of case-2.

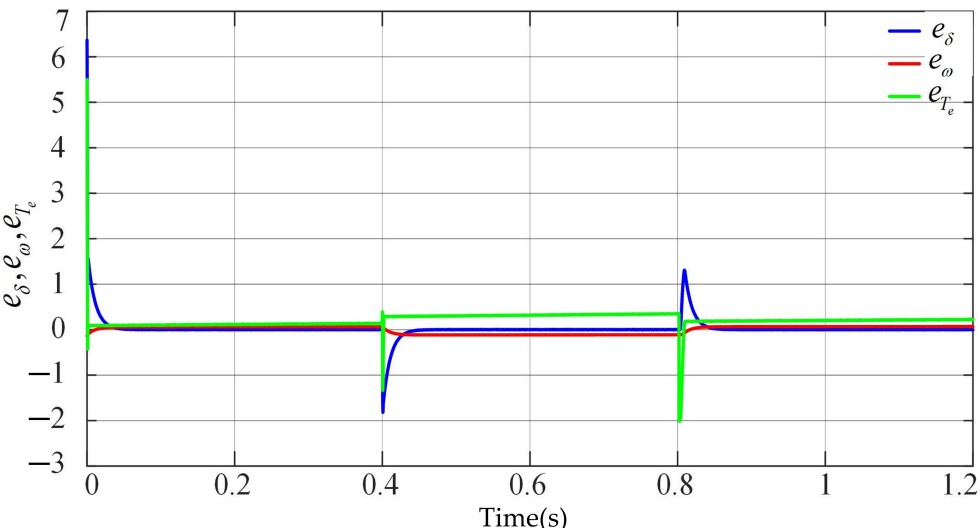

**Figure 9.** System tracking errors $e_\delta$, $e_\omega$, and $e_{T_e}$ in case-1.

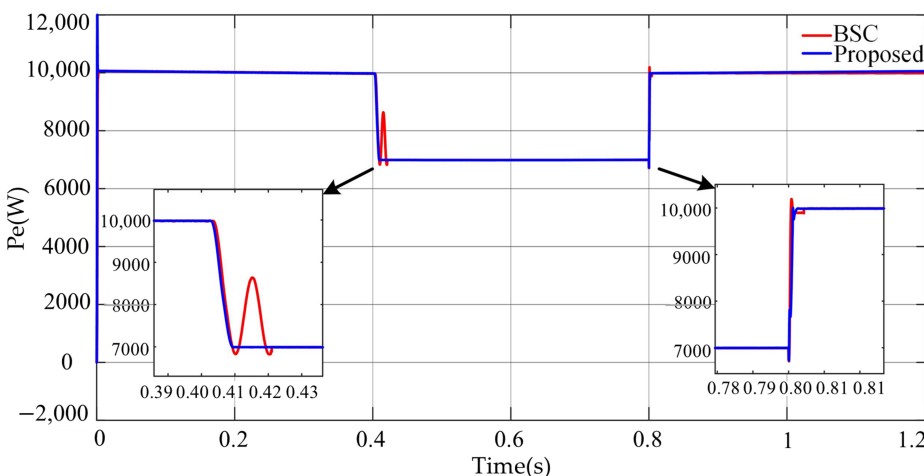

**Figure 10.** Active power variation result of case-2.

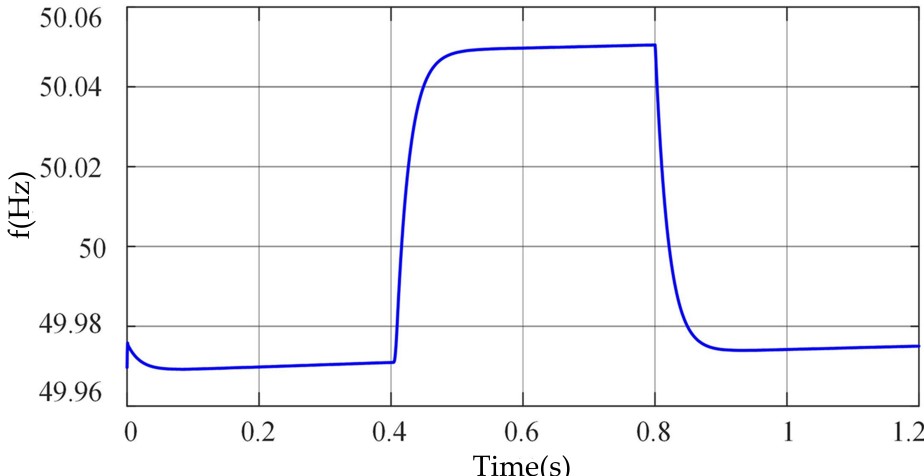

**Figure 11.** Frequency variation result of case-2.

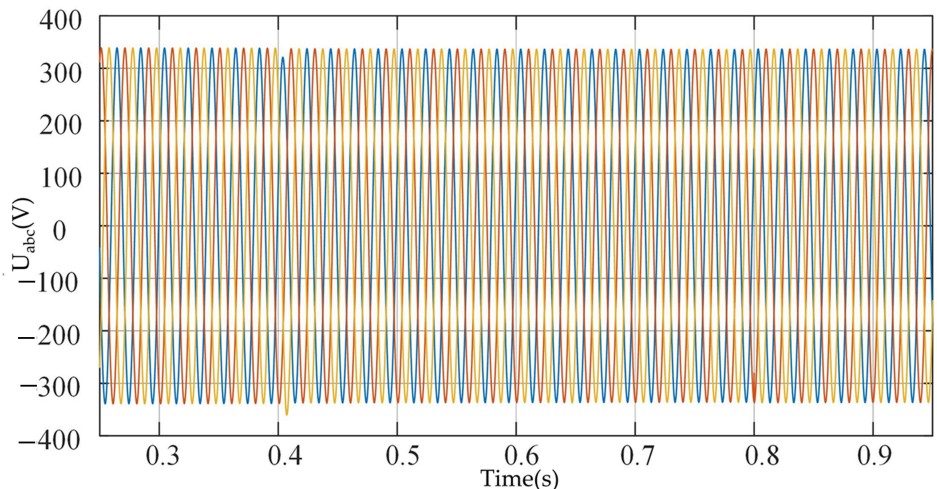

**Figure 12.** Three-phase voltage variation situation result of case-2.

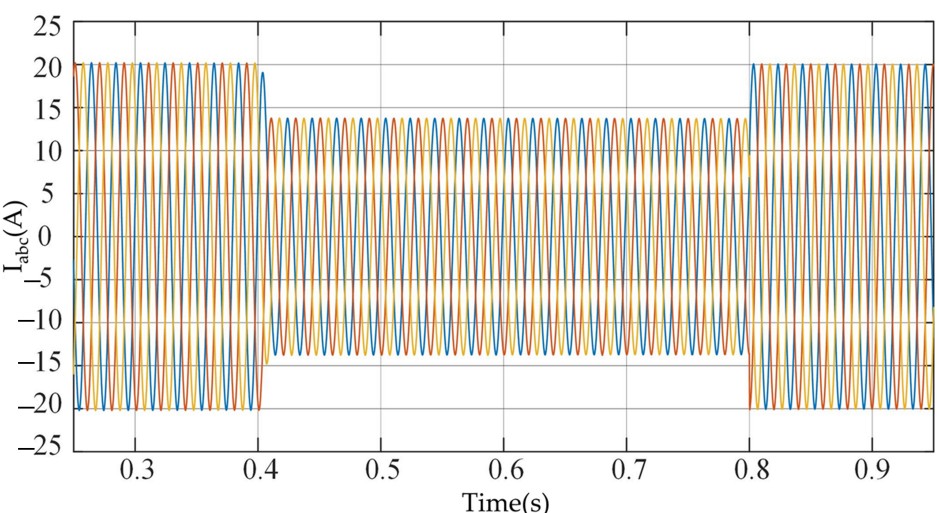

**Figure 13.** Three-phase current variation situation result of case-2.

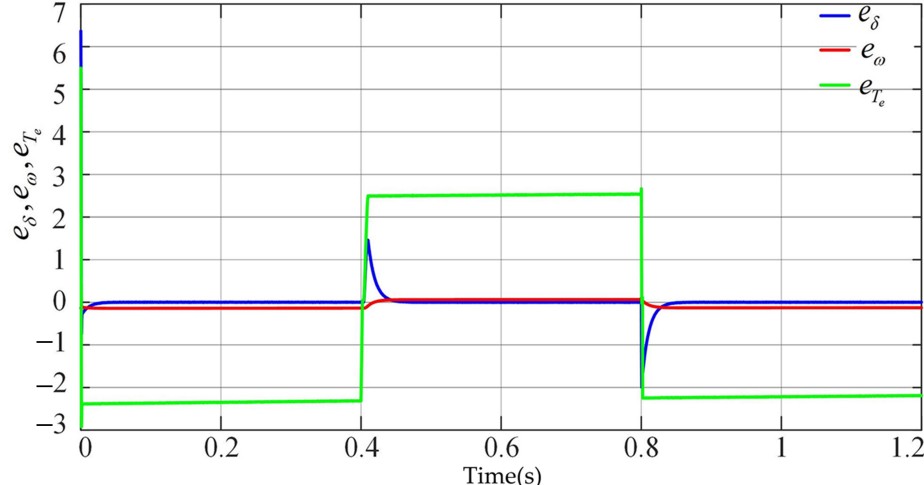

**Figure 14.** System tracking errors $e_\delta$, $e_\omega$, and $e_{T_e}$ of case-2.

### 4.2. Transition Process

The connection and disconnection process also affects the system's output parameters, so testing the controller's robustness in such conditions is necessary to ensure stable operation. Therefore, in this case, the simulation results demonstrate the control performance of the proposed controller during the system transfer from the off-grid to the grid-connected. The system presented in Figure 1 works in off-grid mode first, and the rated power is 10 kW of active power. Because the grid is running correctly and there is no failure condition, the system needs to be connected to the grid, and CB is closed at t = 0.4 s. Due to sudden failure on the grid side at t = 0.8 s, CB opens again. In other words, when t = 0.8 s, the system needs to transfer from grid connection operation mode to island operation mode. The corresponding simulation results are depicted in Figures 15–24.

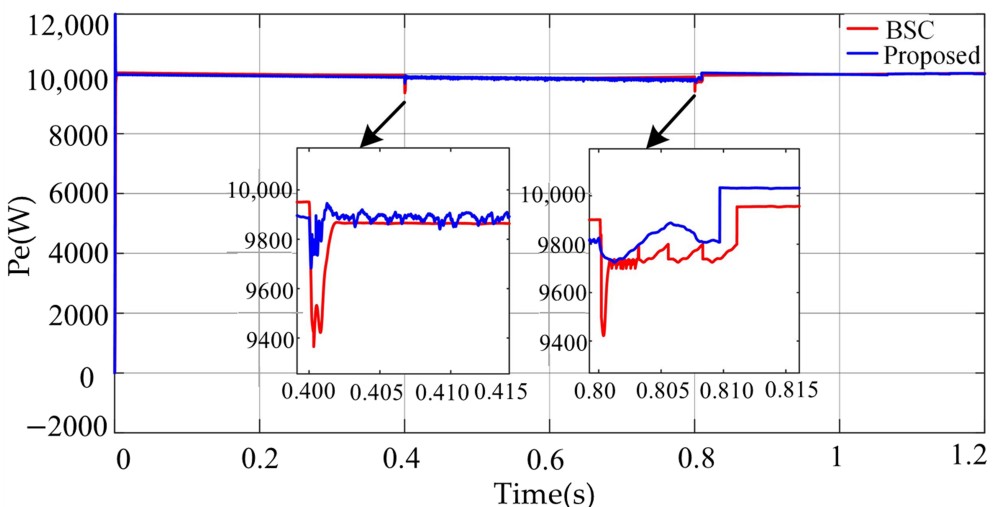

**Figure 15.** Output power result during the transition process.

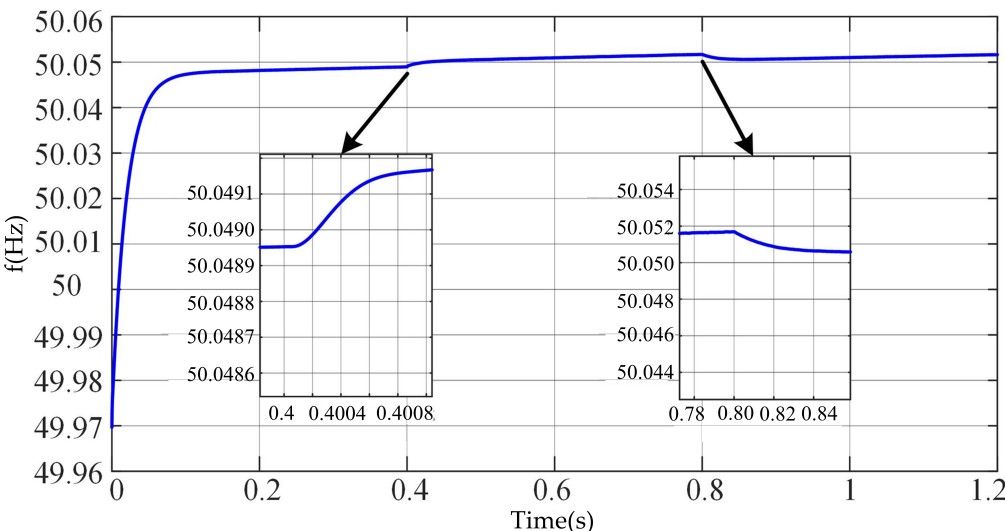

**Figure 16.** System frequency response result during the transition process.

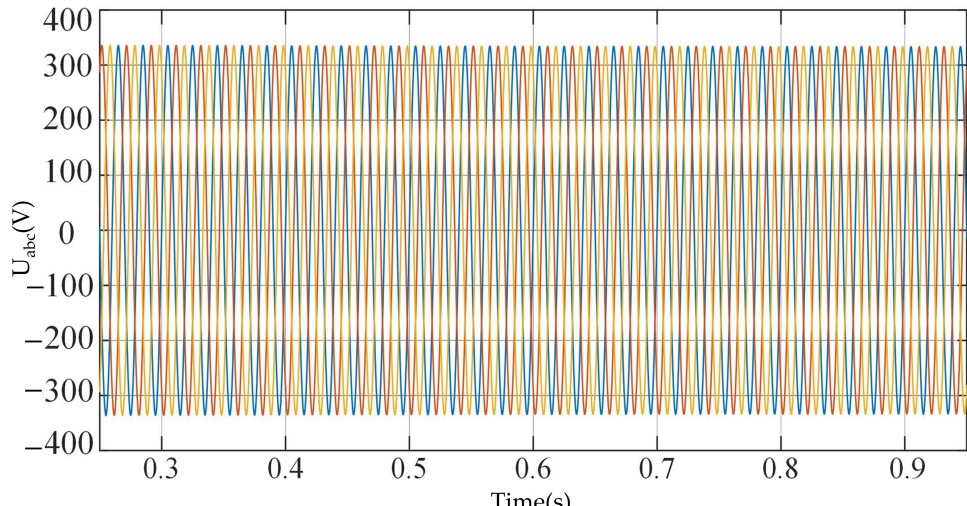

**Figure 17.** Three-phase voltage variation result during the transition process.

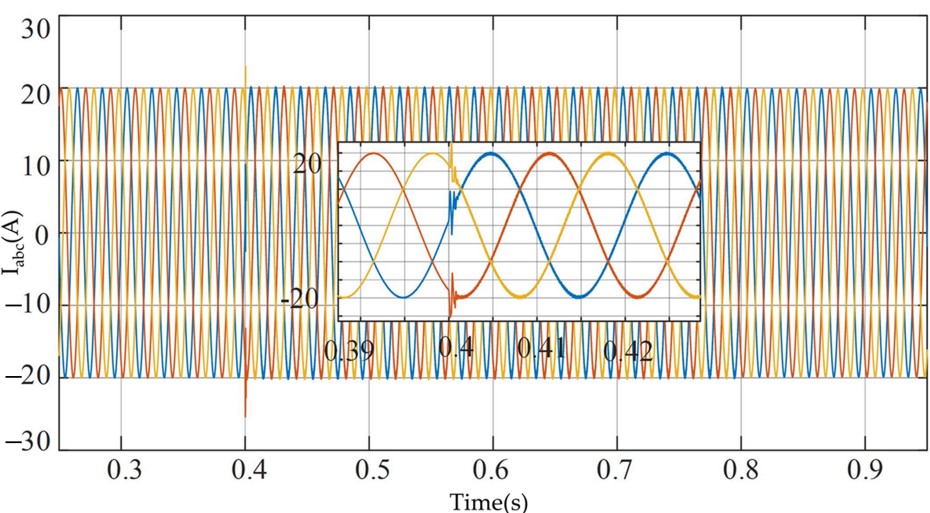

**Figure 18.** Three-phase current variation situation result during the transition process.

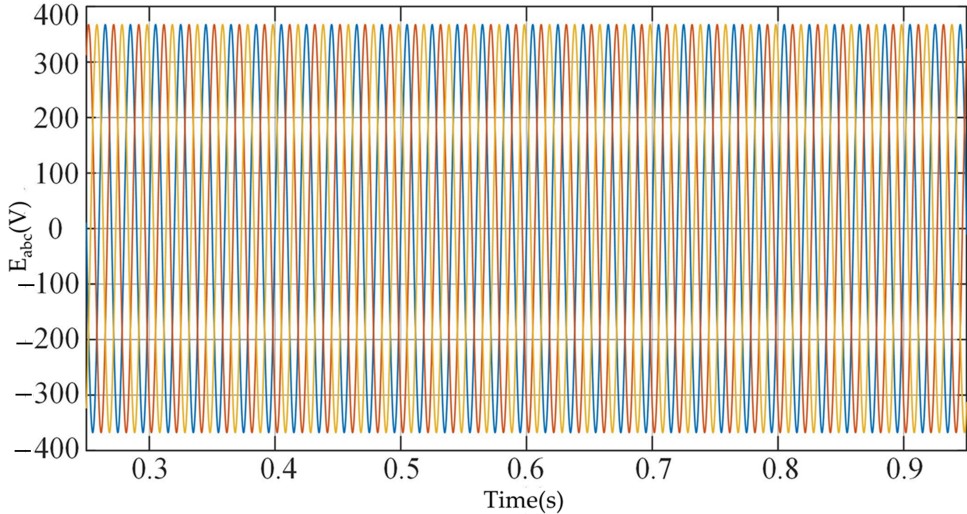

**Figure 19.** Grid voltage variation during the transition process.

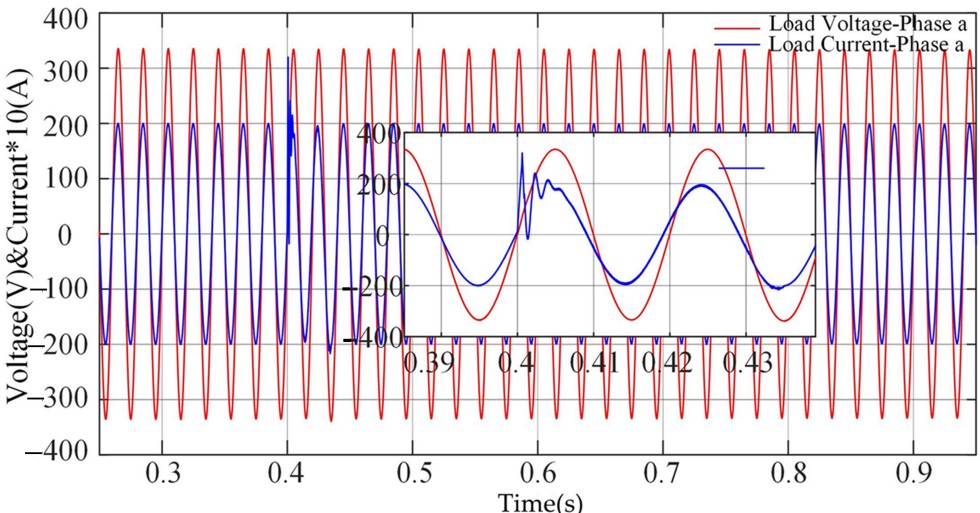

**Figure 20.** Variation of load voltage and current (phase-a) during the transition process.

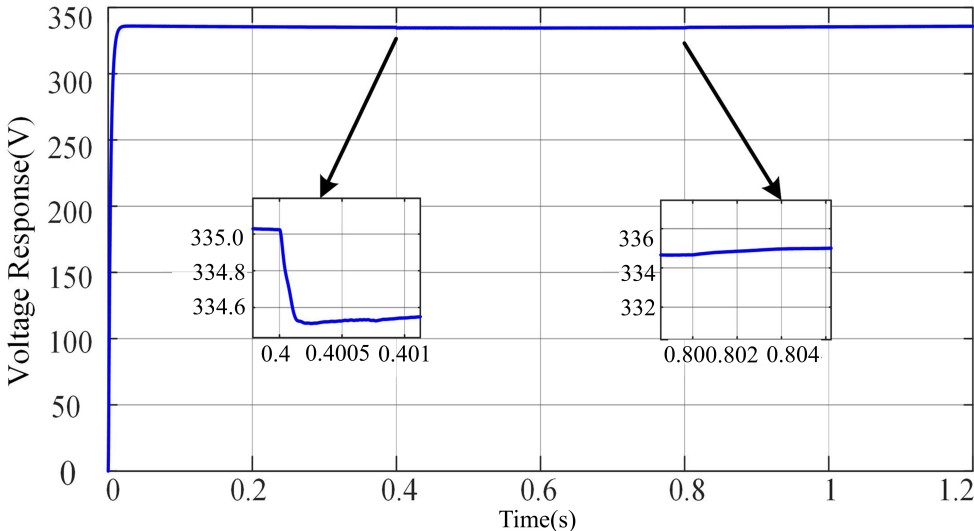

**Figure 21.** Output voltage response during the transition process.

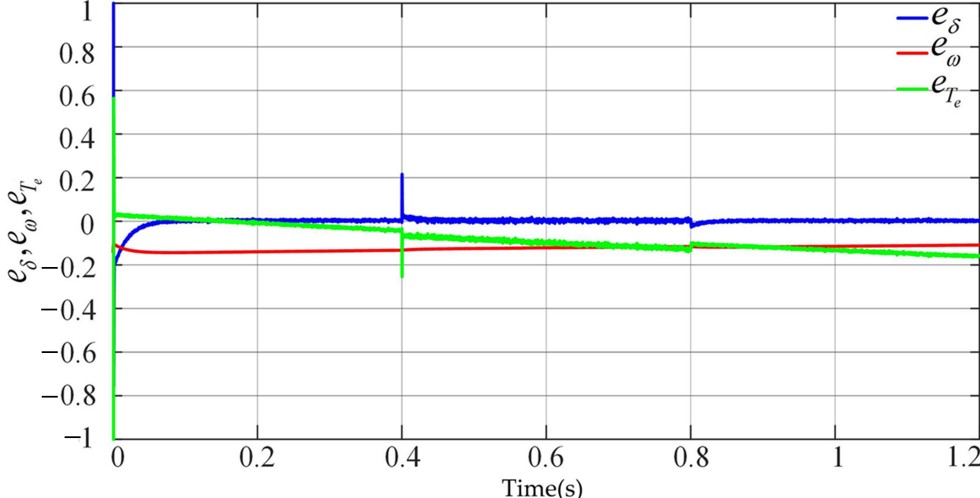

**Figure 22.** System tracking errors $e_\delta$, $e_\omega$, and $e_{T_e}$.

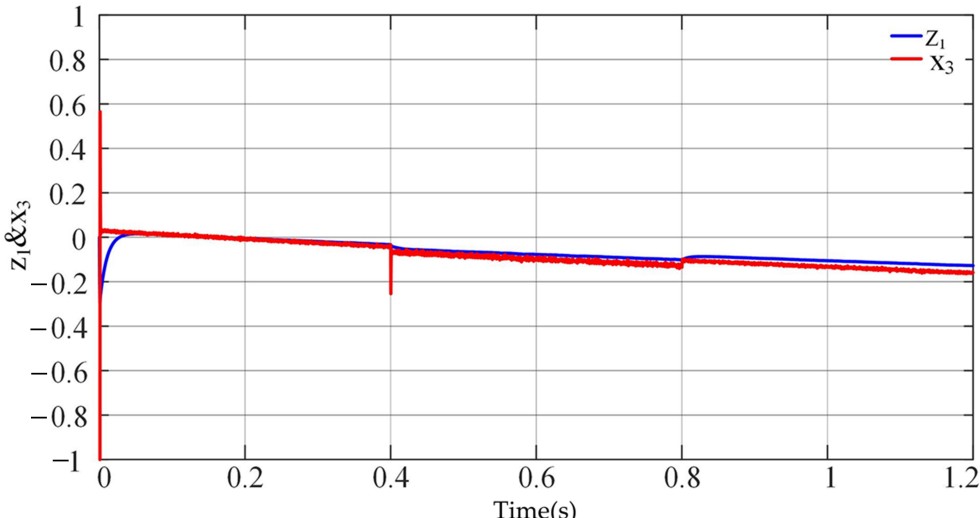

**Figure 23.** System state variable $x_3$ and estimated value $z_1$.

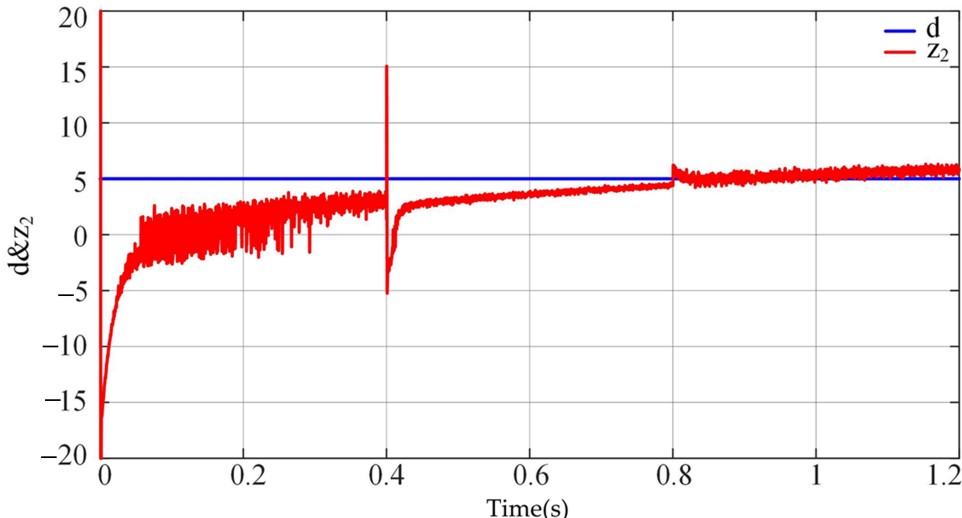

**Figure 24.** Uncertain disturbance $d$ and predicted disturbance $z_2$.

## 5. Discussion

The active power output effects are the main point of discussion and analysis, together with other parameters representing frequency, voltages, and currents, in cases of off-grid and transfer from islanding mode to grid connection modes.

### 5.1. Load Variation Disturbances

#### 5.1.1. Case-1

Figure 5 depicts the active power variation result of simulation under both the proposed control strategy and BSC. It can be understood that the active power can easily and smoothly meet the desired value with the minor amplitude fluctuation at the beginning when t = 0 s, then returned to its target output. Due to adding a 3 kW load at t = 0.4 s, the load variation disturbance causes a sudden power imbalance between the demand and output power; this process leads to a slight fluctuation in the active power to 6871 W, then increases immediately to 10,333 W. Results of the suggested control technique show that the value of the active power is returned to the expected value. At the same time, the BSC oscillates between 6710 W–10350 W at 0.4 s and 8640 W–6830 W for 0.021 s after removing the added load at t = 0.8 s. No changes affect the system output power performance under the proposed control method when the 3 kW load is switched off at t

= 0.8 s. Due to load increasing, the frequency decreases by about 0.075 Hz, as shown in Figure 6, which illustrates the frequency variation during the sudden load increase and decrease simulation result, where it is evident that the system frequency varies between 49.97 Hz and 50.05 Hz; this range remains within an acceptable value. Figures 7 and 8 show the system output voltages and currents in three phases, respectively. The voltage and current waveforms can easily remain unchanged when the system is suddenly loaded under the suggested control technique. Figure 9 shows the tracking errors of the system state variables during the step load change, which are almost zero.

### 5.1.2. Case-2

Figure 10 demonstrates active power variation throughout simulation testing under the proposed control approach and BSC; it is evident that the active power curve of the proposed control strategy can provide good transient responses with no fluctuation when the load is reduced at t = 0.4 s, while the maximum fluctuation of the BSC is between 6830 W–8650 W during the interval 0.4–0.421 s. However, when the load increases again, the active power drops to 6730 W at t = 0.8 s, then immediately climbs to 10,030 W before returning to its steady state under the recommended control mechanism; simultaneously, the oscillation in the case of BSC ranges between 6709.8 W–10,192.4 W. Figure 11 shows the frequency variation result of case-2; according to this result, the system operates at a frequency lower than the reference frequency; when 3 kW is removed, the frequency increases by 0.078 Hz due to load decrease until t = 0.8 s, and when the load is added again the frequency decreases by about 0.025 Hz. Figures 12 and 13 represent the system output voltage and current waveforms, respectively. It is clear that during this case, the three-phase waves of voltage and current also remain unchanged, which is a unique feature of this control approach. Figure 14 represents the system tracking errors of the state variables during the load changing in case-2.

According to the previous simulation experiment results, for VSG working in an off-grid system, under the proposed strategy, the oscillation amplitude of the system output parameters caused by load fluctuation is mostly minor, and the time of restoration to stability is fast, which indicates satisfactory obtained results.

### 5.2. Transition Process

Figure 15 demonstrates the active power output result at the moment of grid connecting for both the suggested control approach and BSC. The curves show that due to the power angle needing to be restored, the active power decreases to 9700 W under the recommended method and 9400 W under BSC at t = 0.4 s, and then quickly returns to the desired value until the system disconnects at t = 0.8 s. The system is exposed to a slight oscillation for a short period and then disappears due to the proposed control mechanism at the moment of disconnecting, while in the case of BSC, the output power drops to 9420 W and then returns to a stable state at the moment of disconnecting. As it can be understood from the simulation result introduced by Figure 15, the proposed controller developed in this paper can offer a more excellent stable operation and provides smoother output power during transient actions than BSC, particularly in the 0.4 s and 0.8 s intervals. The fluctuation range is also smaller than the BSC, and the control performance is more acceptable.

Figure 16 shows the frequency variation during the off-grid and grid connection simulation test. It can be seen that the system frequency is stable and remains close to the reference value with not too much change; this result reflects the performance of the suggested controller for improving system stability. Figures 17–21 represent the system output parameters of three-phase voltage, three-phase current, grid voltage, load voltage and current, and voltage response variation, respectively. It can be noted that the suggested control method's oscillation ranges during the transfer from islanding to grid connecting and vice versa are very small, resulting in a smooth switching process without any disturbing deviations or causing voltage transients. Nevertheless, the three-

phase current waveforms in Figure 18 have a relatively large deviation when the CB is switched on at t = 0.4 s, which fluctuates between 23A and −25.3A and maintains 0.006 s before returning to the desired value by the proposed strategy. When there is a more significant gap between the grid voltage and the VSG output voltage, the inpouring current will also grow, and hence there will be an increase in power variation and system instability. The effect of inpouring current fluctuation on the output power is illustrated in Figure 15. The system tracking errors, state variable $x_3$ with estimated value $z_1$, and uncertain disturbance $d$ with predicted disturbance $z_2$ during this simulation experiment test are shown in Figures 23 and 24, respectively. The system tracking errors of curves in Figure 22 are close to zero, which indicates the stability process. Knowing that, the error $e_{T_e}$ can monitor changes and adapt the system more precisely to achieve a stable operation. Figures 23 and 24 show that the developed ESO can conduct effective dynamic compensation for the uncertain disturbance term and efficient tracking, and this validates what was achieved in the analysis and discussed in Section 3. These results illustrate the extent to which the system under the proposed controller can guarantee stability and faster response, and it has tremendous practical utility. For more verification of the usefulness of this method, Table 3 compares the results obtained in this work with other recently proposed control strategies for improving the microgrid stability objective.

**Table 3.** Comparison of various VSG strategies for stability improvement with the proposed method.

| Strategy | Validation | Microgrid Case | Advantages | Disadvantages |
|---|---|---|---|---|
| Proposed method | MATLAB/ Simulink | -Islanded -Grid connected | -Simple. -Good dynamics response to power step and transition process. -External disturbance is considered. | -Inrush current at the moment of connection. -Absence of experimental test. |
| ATSMC [31] | MATLAB/ Simulink | -Grid connected | -Good tracking performance. -Small frequency deviation during transition process. | -Complicated with many design parameters. -Power deviation during grid connection. |
| OPIC [40] | MATLAB/ Simulink | -Grid connected | -Small RoCoF response to change of system frequency. -Short adjust time. | -Large voltage deviation under sudden load change. |
| ISMBC [30] | MATLAB/ Simulink | -Grid connected | -Simple. -Good tracking performance. -Includes external disturbances. | -Power deviation during grid connection. |
| ACBC [8] | MATLAB/ Simulink | -Islanded -Grid connected | -Includes multi-VSG. -Smooth transition process. -Short adjust time. | -Complicated. -Power deviation during demand change. |
| ATSMBSC [41] | MATLAB/ Simulink | -Grid connected | -Simple. -Good dynamics when changes to islanded mode. -Less power adjust time | -Voltage transients during grid connection. -Large power deviation during grid connection. |
| FIS [42] | Simulink/ Experiment | -Islanded -Grid connected | -Simple. -Includes multi-VSG. -Short adjust time. | -Poor dynamic response to load disturbance. |

## 6. Conclusions

In response to low inertia and damping properties issues in a microgrid system, the system shows a low output impedance, leading to instability, particularly in the transition process. This paper designs a new control method that combines backstepping control with an extended state observer to improve the operational performance of the proposed inverter topology. It enables a seamless transition between off-grid and grid-tied; thereby, the overall control performance of the microgrid will improve. The linear control part, which represents the rotor swing equations, is presented based on the synchronous generator model. Then, the nonlinear controller is constructed, and controller stability is proved using the Lyapunov function.

The simulation results affirm the adequacy of the proposed control technique so that the microgrid system's stability and dynamic performance are verified. Moreover, uncer-

tainties, including external disturbances and tracking errors, are taken into consideration and predicted using the proposed extended state observer with high accuracy performance. Theoretical evaluation shows that the developed controller for the virtual synchronous generator can reduce the power oscillation at the moment of load demand change and provide a seamless transition between off-grid and grid-tied compared with traditional backstepping control, which validates the greatest control performance under the suggested control technique for the microgrid system.

In future work, we will build an experiment platform of a small microgrid, including multi-VSGs, to verify our suggested controller's practical effectiveness.

**Author Contributions:** Conceptualization, S.I.A.H., Y.Z. (Yun Zeng), and J.Q.; methodology, S.I.A.H. and J.Q.; software, S.I.A.H., Y.Z. (Yidong Zou) and D.T.; validation, S.I.A.H. and J.Q.; writing: original draft preparation, S.I.A.H.; writing: review and editing, Y.Z. (Yidong Zou) and D.T.; supervision, Y.Z. (Yun Zeng) and J.Q.; funding acquisition, Y.Z. (Yun Zeng). All authors have read and agreed to the published version of the manuscript.

**Funding:** This research was funded by the National Natural Science Foundation of China, grant numbers (51869007, 52079059).

**Conflicts of Interest:** The authors declare no conflicts of interest.

## Abbreviations

The following abbreviations are used in Table 3, Section 5:

| | |
|---|---|
| ATSMC | Adaptive Terminal Sliding Mode Control |
| OPIC | Optimized Proportional-Integral Controller |
| ISMBC | Integral Sliding Mode Backstepping Control |
| ACBC | Adaptive Command-Filter Backstepping Control |
| ATSMBSC | Adaptive Terminal Sliding Mode Backstepping Control |
| FIS | Fuzzy Inference System |

## Appendix A

The following derivations are used in this manuscript:

$$E = M_f i_f \dot{\theta} \sin\widetilde{\theta} - M_f \frac{di_f}{dt}\cos\widetilde{\theta} \Rightarrow E = \psi_f \omega \sin\widetilde{\theta} \tag{A1}$$

$$P_e = 1.5\psi_f i\omega \cos(\theta - \varphi) = 1.5\psi_f i\omega \cos\delta \tag{A2}$$

$$Q_e = 1.5\psi_f i\omega \sin(\theta - \varphi) = 1.5\psi_f i\omega \sin\delta \tag{A3}$$

$$i_{abc} = \begin{bmatrix} i_a \\ i_b \\ i_c \end{bmatrix} = i\begin{bmatrix} \cos(\varphi) \\ \cos(\varphi - \frac{2\pi}{3}) \\ \cos(\varphi - \frac{4\pi}{3}) \end{bmatrix} \tag{A4}$$

$$i_{abc} = \begin{bmatrix} e_a \\ e_b \\ e_c \end{bmatrix} = M_f i_f \dot{\theta}\sin\widetilde{\theta} = \psi_f\omega\sin\widetilde{\theta} = \psi_f\omega\begin{bmatrix} \sin(\theta) \\ \sin(\theta - \frac{2\pi}{3}) \\ \sin(\theta - \frac{4\pi}{3}) \end{bmatrix} \tag{A5}$$

$$u_1 = \frac{1}{Ks}\{V_{ref} - V_{act}\} = \frac{1}{Ks}\{V_{rat} - D_q(Q_{ref} - Q_e) - V_{act}\} \tag{A6}$$

$$\psi_f(1 + t_e s) = t_e u_2 \Rightarrow \dot{\psi}_f = \frac{k_e}{t_e}u_2 - \frac{\psi_f}{T_E} \tag{A7}$$

The observer dynamics error:

$$\begin{cases} e_1 = z_1 - x_3, \\ e_2 = z_2 - d. \end{cases} \tag{A8}$$

$$\begin{cases} \dot{e}_1 = e_2 - \beta_1 e_1, \\ \dot{e}_2 = -\beta_2 fal(e_1, \alpha, \tau) - \dot{d}. \end{cases} \tag{A9}$$

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
