# Peer review of "Extended State Observer Based-Backstepping Control for Virtual Synchronous Generator"

_electronics, doi:10.3390/electronics11192988_

Round 1

Reviewer 1 Report

Review results

The paper "Extended State Observer Based-Backstepping Control for Virtual Synchronous Generator" presented by the authors presents interesting results that can be useful to those interested in the control of power electronics generators. They offer a nonlinear control strategy for VSG with uncertain disturbance proposed in this paper to enhance the system stability in the islanded, grid-connected, and transition modes. The paper is properly organized and carefully explained. The results presented by the authors are clearly explained, as well as the methodology and approach they used to obtain such results. For this reason, I could recommend this paper for its publication, but I believe that a minor revision is needed in order to correct some writing and punctuation issues.

Additionally:

1. On line 153, the authors defined x_r as the reference value for x_1, but a relationship with x_2 is not mentioned, is it the same? The authors must clarify.

2. On line 181, the authors mention that selected k's are to achieve rapid stability, Could the authors show a figure showing the convenience of these values between all the other possible options to justify its election?

Author Response

September 11th, 2022

Manuscript ID: Electronics-1901783

Response to Reviewer

Dear Reviewer,

Thank you for your kind comments on our manuscript entitled “Extended State Observer Based-Backstepping Control for Virtual Synchronous Generator” to help improve the quality of our manuscript. We have carefully revised the manuscript according to your comments, and we addressed these issues in the highlighted version (track changes, marked by yellow colour). Below we explain our responses to your suggestions.

The paper has been carefully reviewed, some writing and punctuation issues have been corrected, and changes are addressed in a new version.

Additionally:

Point 1: On line 153, the authors defined xr as the reference value for x1, but a relationship with x2 is not mentioned, is it the same? The authors must clarify.

Response 1: Thank you for your kind comment, and we are pleased to explain it. x2 is the derivative value of x1, and x1 represents the rate change of δ. In other words, x1 is proportional to x2.

Point 2: On line 181, the authors mention that selected k's are to achieve rapid stability, Could the authors show a figure showing the convenience of these values between all the other possible options to justify its election?

Response 2: Thank you again for your comment and inquiry, the K values you referred to in this comment were chosen by try and error to achieve the stability indicated in Section 3, Equations (20,27 and 32). The selected values are reflected in the results obtained in this paper.

Thank you very much for your time!

Sincerely,

Shamseldeen Ismail Abdallah Haroon

Reviewer 2 Report

The work done by the authors titled "Extended State Observer Based-Back stepping Control for Virtual Synchronous Generator" is very appreciated. it presents The penetration of distributed generators(DGs)-based power electronic devices lead to1 low inertia and damping properties of the modern power grid. As a result, the system becomes2 more susceptible to disruption and instability, particularly when the power demand changes during3 critical loads or the system needs to switch from standalone to a grid-connected operation mode or4 vice versa. Developing a robust controller to deal with these transient cases is a real challenge. The5 inverter control method via the virtual synchronous generator (VSG) control method is a better way6 to supply the system’s inertia and damping features to boost system stability. Therefore, a nonlinear7 control strategy for VSG with uncertain disturbance is proposed in this paper to enhance the system8 stability in the islanded, grid-connected, and transition modes. Firstly, the mechanical equations for9 VSG’s rotor, which include virtual inertia and damping coefficient, are presented, and the matching10 mathematical model is produced. However the following improvements are needed on it

1. The resolution of the figures is very low, they must be improved. So that the scale and legends are readable to the readers. 

2. There is a lack of explanation about the results

3. Arrange the paper in IMRAD format. Merge some sections when needed

4. Keep a discussion section before conclusion, and compare the obtained results with the literature at least 10 references with citations. If possible produce a table for comparative assessment, so that the readers are more impressed on this work

5. Table -1, covers only few parameters, what about the other parameters.

6. You derived more equations as like lectures class or students presentation in exam. Please use required equations only and explain why and how that equation is involved in your proposed research?. Remaining derivation, keep as a supplementary file or Appendix

7. In results explain how the changes are obtained. Ex. Increase or decrease in graph and Why?

8. The references are covered upto 2021 only? No work or related work is done in 2022?

9. What about the experimental validation, If your institute supports grant why not you conducted experiment. Without experiment, how you justify this results in your institute? is this acceptable?

10. Are these assumptions 1&2, reflected in your results? where and How?

Justify above. All the best. 

Author Response

September 11th, 2022

Manuscript ID: Electronics-1901783

Response to Reviewer

Dear Reviewer,

Thank you for your kind comments on our manuscript entitled “Extended State Observer Based-Backstepping Control for Virtual Synchronous Generator” to help improve the quality of our manuscript. We have carefully revised the manuscript according to your comments, and we addressed these issues in the highlighted version (track changes, marked by yellow colour). Below we explain our responses to your suggestions.

Thank you very much for your time!

Sincerely,

Shamseldeen Ismail Abdallah Haroon

Round 2

Reviewer 2 Report

The revisions are up to the mark and may be recommended for publication.